# New insights on the modeling of the molecular mechanisms underlying neural maps alignment in the midbrain

**Elise Laura Savier**[1]*, **James Dunbar**[2,3], **Kyle Cheung**[2], **Michael Reber**[2,3,4,5,6]*

[1]Department of Biology and Psychology, University of Virginia, Charlottesville, United States; [2]Donald K. Johnson Eye Institute, Krembil Research Institute, University Health Network, Toronto, Canada; [3]Laboratory of Medicine and Pathobiology, University of Toronto, Toronto, Canada; [4]Ophthalmology and Vision Sciences, University of Toronto, Toronto, Canada; [5]Cell and System Biology, University of Toronto, Toronto, Canada; [6]CNRS UPR 3212, University of Strasbourg, Strasbourg, France

**Abstract** We previously identified and modeled a principle of visual map alignment in the midbrain involving the mapping of the retinal projections and concurrent transposition of retinal guidance cues into the superior colliculus providing positional information for the organization of cortical V1 projections onto the retinal map (Savier et al., 2017). This principle relies on mechanisms involving Epha/Efna signaling, correlated neuronal activity and axon competition. Here, using the 3-step map alignment computational model, we predict and validate in vivo the visual mapping defects in a well-characterized mouse model. Our results challenge previous hypotheses and provide an alternative, although complementary, explanation for the phenotype observed. In addition, we propose a new quantification method to assess the degree of alignment and organization between maps, allowing inter-model comparisons. This work generalizes the validity and robustness of the 3-step map alignment algorithm as a predictive tool and confirms the basic mechanisms of visual map organization.

**\*For correspondence:**
els6f@virginia.edu (ELS);
michael.reber@uhnresearch.ca
(MR)

**Competing interests:** The authors declare that no competing interests exist.

## Introduction

Understanding and modeling the mechanisms of neural circuits formation in the brain has been a challenging subject in fundamental neurobiology for decades. One of the most studied biological models to investigate the formation and function of sensory connectivity -or sensory maps- from both experimental and theoretical standpoints is the superior colliculus (SC), an evolutionary conserved structure located in the midbrain (*Cang and Feldheim, 2013*; *Basso and May, 2017*). In most vertebrates, the SC is the premier brain center for integrating sensory inputs from visual, auditory and somatosensory modalities distributed in different interacting laminae. This structure is a key node in the network of brain areas responsible for controlling the location of attention and even decision-making (*Basso and May, 2017*). Visual inputs in the superficial layers of the SC correspond to the organized projections from the retinal ganglion cells (RGCs) in the retina (the retino-collicular (RC) projections) and from layer V neurons in the primary visual cortex V1 (the cortico-collicular [CC] projections). Both projections form visual maps that must be aligned and in register to allow efficient detection of visual stimuli (*Basso and May, 2017*; *Liang et al., 2015*; *Zhao et al., 2014*).

Although many studies have focused on the mechanisms involved in the formation of the RC map, little is known about how the RC and CC maps are aligned during development. Previous studies on RC mapping have revealed the involvement of gradients of Eph tyrosine kinases receptors (Eph) and their cognate membrane-bound ligands, the ephrins (Efn), together with competition

between axons for collicular space and correlated neuronal activity in the form of spontaneous waves (*Ackman and Crair, 2014*; *Cang and Feldheim, 2013*; *Triplett et al., 2011*). To gain insight into the molecular mechanisms involved in map formation, mouse genetic models were generated, in which the expression of members of the Eph and ephrin family was manipulated. The mapping mechanisms of the nasal-temporal axis of the retina onto the rostral-caudal axis of the SC involving Epha/Efna signaling have been, by far, the most studied experimentally (*Feldheim et al., 1998*; *Brown et al., 2000*; *Feldheim et al., 2000*; *Reber et al., 2004*; *Rashid et al., 2005*; *Pfeiffenberger et al., 2006*; *Triplett et al., 2009*; *Bevins et al., 2011*; *Suetterlin and Drescher, 2014*; *Owens et al., 2015*; *Savier et al., 2017*; *Savier and Reber, 2018*). Several hypotheses have been made to account for the phenotypes observed in mouse genetic models and this extensive body of work has been used to generate computational approaches which attempt to replicate experimental findings (*Honda, 2003*; *Reber et al., 2004*; *Honda, 2004*; *Koulakov and Tsigankov, 2004*; *Goodhill and Xu, 2005*; *Willshaw, 2006*; *Tsigankov and Koulakov, 2006*; *Tsigankov and Koulakov, 2010*; *Triplett et al., 2011*; *Simpson and Goodhill, 2011*; *Grimbert and Cang, 2012*; *Sterratt and Hjorth, 2013*; *Willshaw et al., 2014*; *Hjorth et al., 2015*; *Owens et al., 2015*; *Savier et al., 2017*). However, until recently, these models have not been able to explain how collicular visual maps are aligned during development.

Interestingly, the formation of the RC map occurs prior to the establishment of the CC map. Other studies have shown that the existence of the RC map is necessary for the formation of the CC map, suggesting an interdependence of the two mechanisms (*Khachab and Bruce, 1999*; *Rhoades et al., 1985*; *Triplett et al., 2009*). Similar observations have been made in other part of the visual system (*Shanks et al., 2016*). Another piece of evidence came from the study of map alignment in the Isl2-Epha3KI, one of the best characterized mutant in the field. In these mutants, the Epha3 receptor is ectopically expressed in 50% of RGCs, leading to a duplication of the RC map (*Brown et al., 2000*). Strikingly, a full duplication of the CC map is also observed in the Isl2$^{Epha3/Epha3}$ homozygous mutants, which display a normal retinotopy in the visual cortex (*Triplett et al., 2009*). The authors concluded that a retinal-matching mechanism involving spontaneous correlated activity in the retina instructs CC projections and alignment onto the RC map. Alternative explanations can also be suggested. For instance, the ectopic expression of Epha3 specifically in the retina may alter the expression of other members of Epha/Efna throughout the visual system disrupting maps formation and alignment. In another example, molecular cues originating from the retina could be carried over to the colliculus and provide mapping/alignment information to ingrowing cortical axons, as suggested earlier (*Savier et al., 2017*). This latter hypothesis is in line with a retinal-matching mechanism inferred by Triplett and collaborators (*Triplett et al., 2009*). A similar mechanism of guidance cues transportation has recently been demonstrated for axon guidance at the optic chiasm in the mouse visual system (*Peng et al., 2018*).

To gain insight into the implication of molecular guidance cues in the retina in the alignment of visual maps in the SC, we have characterized a new mutant, the Isl2-Efna3KI, which over-expresses the ligand Efna3 in 50% of the RGCs (*Savier et al., 2017*). To our surprise, this mutant does not display any defect in the formation of the RC map, however the subsequent CC map is duplicated. This led us to conclude that molecular guidance cues expressed in the retina are implicated in the formation of the CC map (*Savier et al., 2017*). To simulate this mechanism in silico, we generated the 3-step map alignment model, based on the Koulakov model (*Koulakov and Tsigankov, 2004*; *Tsigankov and Koulakov, 2010*; *Tsigankov and Koulakov, 2006*). Many different algorithms have been generated to model RC mapping controlled by Eph/Efn. Recently, *Hjorth et al., 2015* developed a pipeline which allowed for systematic testing of the currently available models and revealed that most cannot reproduce all nuances of experimental findings (Isl2-Epha3KI, EfnaKOs and Math5KO). Among those tested, the most faithful was the Koulakov model, which had recently been extended to explain the variability in the phenotypes observed in a particular mouse model, the Isl2-Epha3KI (*Owens et al., 2015*).

The 3-step map alignment model simulates the formation of the RC map and, for the first time, the subsequent formation and alignment of the CC map along the rostral-caudal axis of the SC based on experimental and mechanistic evidence (*Savier et al., 2017*; *Savier and Reber, 2018*).

This algorithm predicts normal wild-type (WT) mapping as well as the map alignment defects observed in the Isl2-Efna3KI animals (*Savier et al., 2017*). However, whether it also simulates mapping abnormalities observed in other Epha/Efna mutants, and particularly in the Isl2-Epha3KI animal model, is unknown. Here we demonstrate that the 3-step map alignment algorithm accurately simulates both RC and CC mapping defects in Isl2-Epha3KI mutants (*Brown et al., 2000*; *Triplett et al., 2009*). Our results strongly suggest that the mechanism underlying the subsequent duplication of the CC projection corresponds to a redistribution of retinal molecular cues, the Efnas, into the SC. The retinal Efnas, provided by the incoming retinal axons within the SC, act together with correlated activity to instruct CC alignment onto the RC map (*Savier et al., 2017*; *Triplett et al., 2009*). We further confirmed and validated the predictions of the algorithm by quantitative in vivo map analysis in both heterozygous and homozygous Isl2-Epha3KI animals. Moreover, a new implementation of the algorithm generates indexes providing a qualitative measure of map organization and allowing comparison of visual map layouts between biological models. Together with our previous work (*Savier et al., 2017*), these data confirm the validity and robustness of our algorithm and reinforces the underlying principle of visual map formation and alignment in the midbrain, where the layout of the dominant RC map specifies the alignment of the CC map through spontaneous correlated activity and transposed retinal Efnas. Such principle dictates the optimal functioning of the system by allowing fine adjustments which compensate for intrinsic variability -or stochasticity- of sensory maps formation.

## Results

### Normal Ephas/Efnas expression in Isl2$^{Epha3/Epha3}$ retinas, SC, and V1 cortex

Retinal expression of *Epha* receptors in Isl2-Epha3KI animals (in which *Epha3* was targeted into the *Isl2* locus) has been extensively characterized (*Reber et al., 2004*). However, whether EPHA3 ectopic expression in Isl2-positive (Isl2(+)) RGCs affects retinal *Efna* co-expression is unknown. We performed a quantitative analysis of retinal *Efna* transcripts in P1/P2 nasal, central, and temporal acutely isolated RGCs (*Savier et al., 2017*), revealing similar expression levels between Isl2$^{Epha3/Epha3}$ and WT littermates. The graded expression of *Efna2* and *Efna5* running form high-nasal to low-temporal is preserved in Isl2$^{Epha3/Epha3}$ animals, whereas the expression of *Efna3* as well as *Epha3* is homogeneous (*Figure 1A – Figure 1—source data 1A*). Transcript analyses in V1 and SC (*Figure 1B – Figure 1—source data 1B*) revealed similar levels of *Efna2/a3/a5* and *Epha4/a7* receptors in Isl2$^{Epha3/Epha3}$ compared to WT littermates at P7 excluding indirect effects on mapping due to local changes of *Ephas/Efnas* gene expression.

### Simulation of the Isl2-Epha3KI mutants retino- and cortico-collicular mapping

Our data above, together with previous work (*Reber et al., 2004*; *Savier et al., 2017*), showed that ectopic expression of Epha3 or Efna3 in Isl2(+) RGCs does not affect endogenous Efnas nor Ephas expression within the retina, SC, and V1. We can reasonably assume that the equations modeling retinal, collicular and cortical Efna gradients as well as cortical Epha gradients related to the Isl2-Epha3KI genotypes are equivalent to the equations previously characterized (measured experimentally or estimated) in WT animals (*Reber et al., 2004*; *Cang et al., 2005*; *Tsigankov and Koulakov, 2006*; *Tsigankov and Koulakov, 2010*; *Bevins et al., 2011*; *Savier et al., 2017*). Measured gradients of Epha receptors (R $_{Epha}$) in RGCs along the nasal-temporal axis (x), R $_{Epha}$ (x)$^{retina}$ (*Figure 2A, G,M*; *Brown et al., 2000*; *Reber et al., 2004*) are modeled by two equations, one corresponding to Isl2(+) RGCs expressing WT levels of Ephas + Epha3 (in *Figure 2G,M*) and the second corresponding to Isl2-negative (Isl2(-)) RGCs expressing only WT levels of Ephas (see Materials and methods for numerical values).

The projections of one hundred RGCs onto the SC, the RC map, were simulated by the 3-step map alignment model ($10^7$ iterations per run, n = 20 runs) for the WT, Isl2$^{Epha3/+}$ and Isl2$^{Epha3/Epha3}$ genotypes (*Figure 2E,K,Q*). RGCs axons/growth cones carrying gradients of Epha receptors are repelled by complementary gradients of collicular Efna (forward signaling) (*Figure 2B,H,N*), while correlated activity attracts projections originating from similar locations through pair-wise attraction,

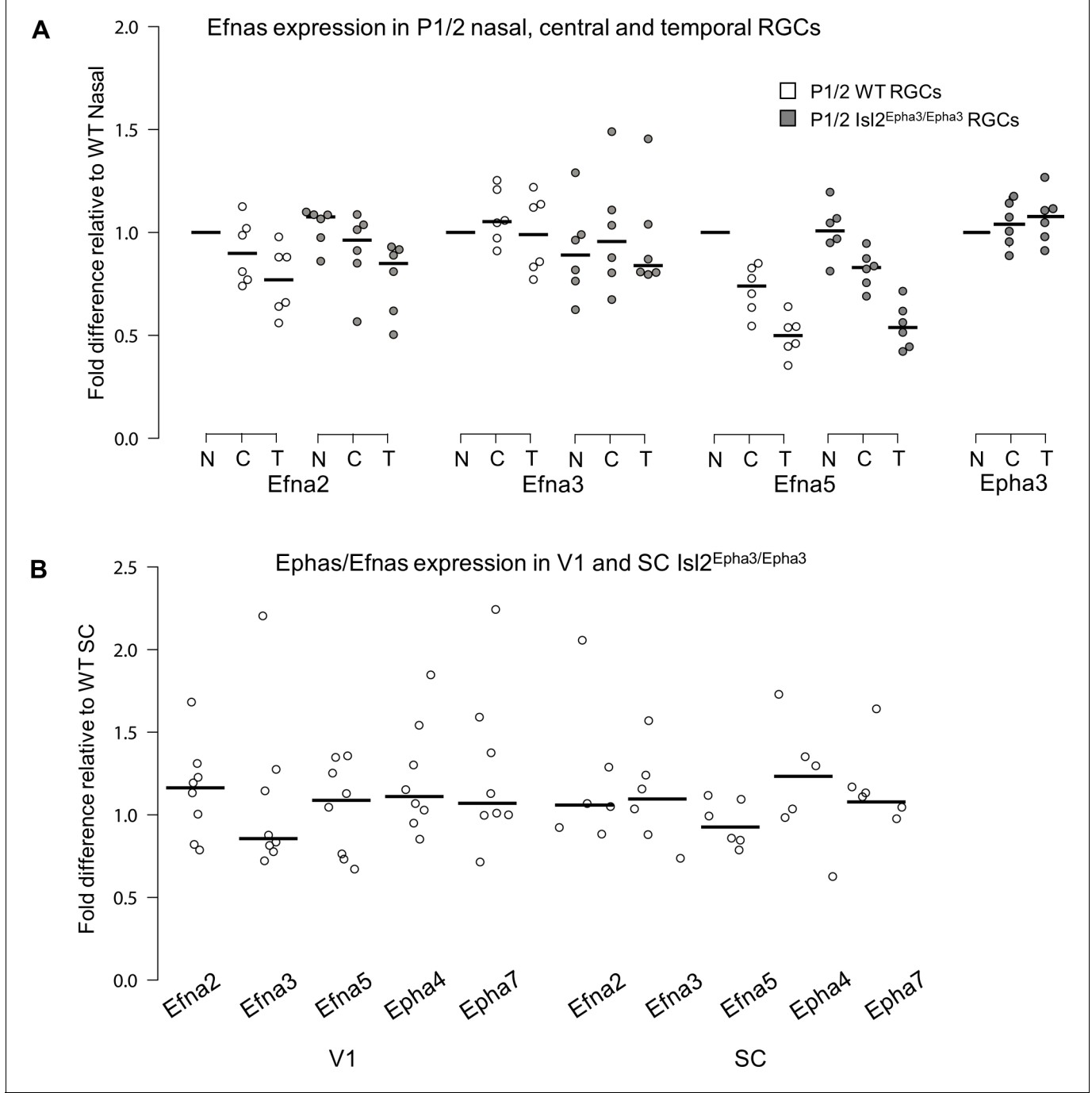

**Figure 1.** Dot plots representing Ephas/Efnas expression in Isl2$^{Epha3/Epha3}$ retinas, V1 cortex and SC. (**A**) Median Efna2/a3/a5 expression levels (relative to wild-type nasal expression) in P1/2 wild-type (WT - white) and Isl2$^{Epha3/Epha3}$ (gray) acutely isolated RGCs from nasal (**N**), central (**C**) and temporal (**T**) retinas (WT, Isl2$^{Epha3/Epha3}$, n = 6 animals, 12 retinas), Two-way ANOVA without replication: Efna2 x genotype: $F_{(1,2)}$ = 3.72 < $F_{crit.}$=18.5, p=0.19; Efna3 x genotype : $F_{(1, 2)}$=11.13 < $F_{crit.}$=18.5, p=0.07; Efna5 x genotype: $F_{(1, 2)}$=3.58 < $F_{crit.}$=18.5, p=0.20. (**B**) Median Efna2/a3/a5 ligands and Epha4/a7 receptors expression levels (relative to WT expression levels) in Isl2$^{Epha3/Epha3}$ V1 (WT n = 5 animals, Isl2$^{Epha3/Epha3}$ n = 8 animals; variables are normally distributed, one sample t-test: Efna2: p=0.29; Efna3: p=0.43; Efna5: p=0.42; Epha4: p=0.07; Epha7: p=0.54) and SC (WT n = 5 animals, Isl2$^{Epha3/Epha3}$ n = 6 animals; variables are normally distributed, one sample t-test: Efna2: p=0.20; Efna3: p=0.65; Efna5: p=0.71; Epha4: p=0.11; Epha7: p=0.17). qPCRs were repeated three times with duplicates for each sample.

The online version of this article includes the following source data for figure 1:

**Source data 1.** Data for *Figure 1A*: Expression levels, relative to wild-type, of retinal Efna in Isl2$^{Epha3/Epha3}$ animals.

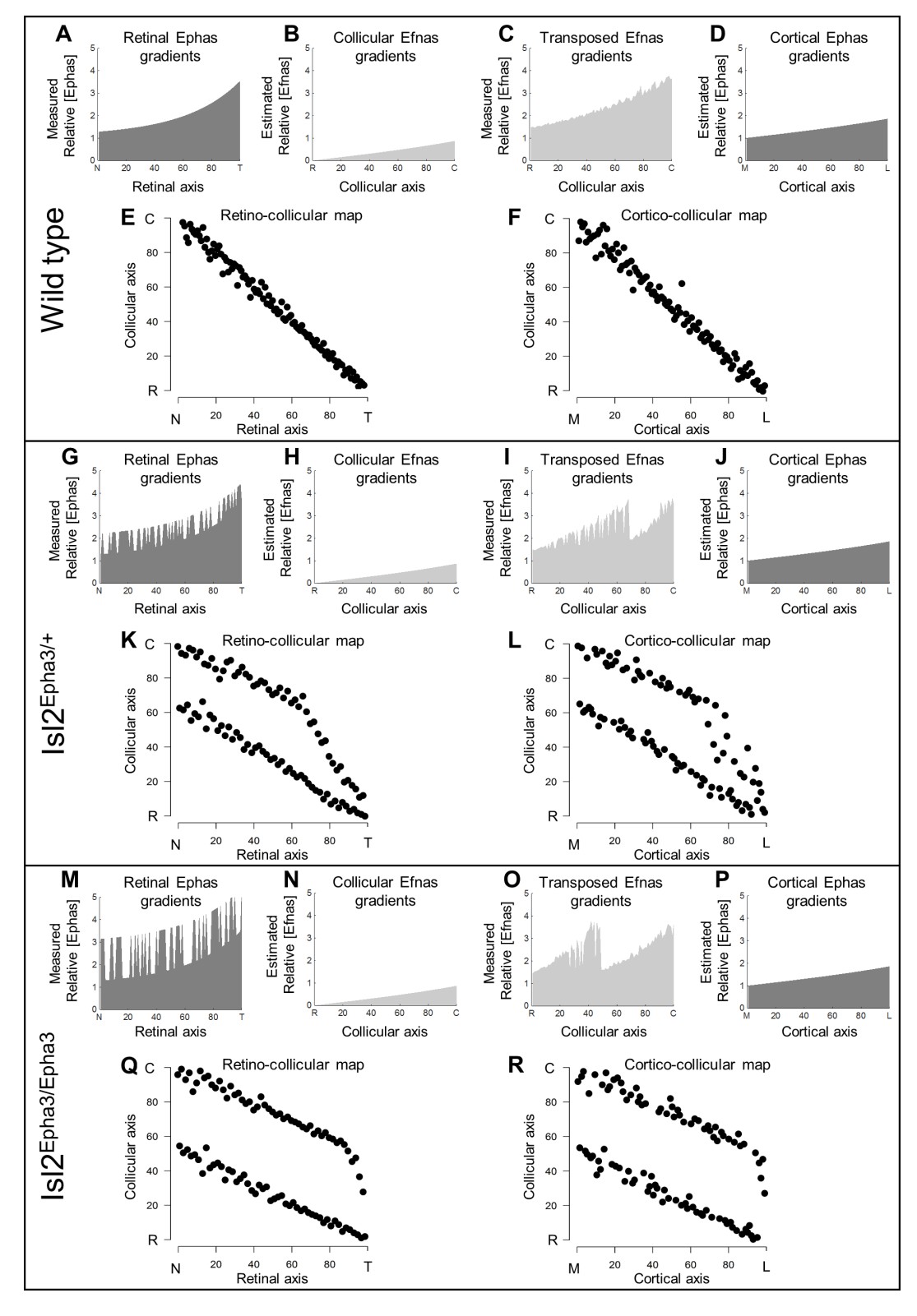

**Figure 2.** Simulations of retino- and cortico-collicular mapping in Isl2-Epha3KI animals. (A, G, M) Representation of measured retinal Epha gradients along the nasal-temporal (NT) axis in WT (A), Isl2^Epha3/+ (G) and Isl2^Epha3/Epha3 (M) animals (see Materials and methods and *Table 1* for equations). (B, H, N) Representation of the estimated collicular Efna gradients along the rostral-caudal (RC) axis in WT (B), Isl2^Epha3/+ (H) and Isl2^Epha3/Epha3 (N) animals (see Materials and methods and *Table 1* for equations). (C, I, O) Representation of the transposed retinal Efna gradients into the SC along the RC axis

*Figure 2 continued on next page*

*Figure 2 continued*

in WT (C), Isl2^Epha3/+ (I) and Isl2^Epha3/Epha3 (O) animals (see Materials and methods and *Table 1* for equations). (D, J, P) Representation of the estimated cortical Epha gradients along the medial-lateral (ML) axis in V1 in WT (D), Isl2^Epha3/+ (J) and Isl2^Epha3/Epha3 (P) animals (see Materials and methods and *Table 1* for equations). (E, K, Q) Simulated RC map in in WT (E), Isl2^Epha3/+ (K) and Isl2^Epha3/Epha3 (Q) animals generated by the 3-step map alignment algorithm (representative of n = 20 runs). (F, L, R) Simulated cortico-collicular map in WT (F), Isl2^Epha3/+ (L) and Isl2^Epha3/Epha3 (R) animals generated by the 3-step map alignment algorithm (representative of n = 20 runs). Abbreviations: N, nasal; T, temporal; R, rostral; C, caudal; M, medial; L, lateral.

which is inversely proportional to the distance between two RGC/V1 neurons. Results showed a continuous, single map for WT (*Figure 2E*), as expected from previous experimental findings and theoretical modeling (*Brown et al., 2000*; *Reber et al., 2004*; *Savier et al., 2017*; *Bevins et al., 2011*). For the Isl2^Epha3/+, partially duplicated RC mapping is modeled, together with the presence of a collapse point for temporal-most RGCs (*Figure 2K*) as extensively observed experimentally (*Brown et al., 2000*; *Reber et al., 2004*; *Bevins et al., 2011*) and theoretically (*Reber et al., 2004*; *Willshaw, 2006*; *Tsigankov and Koulakov, 2010*; *Simpson and Goodhill, 2011*). For Isl2^Epha3/Epha3, the RC map is fully duplicated, similarly to previous experimental and theoretical findings (*Figure 2Q*; *Brown et al., 2000*; *Reber et al., 2004*; *Willshaw, 2006*; *Tsigankov and Koulakov, 2010*; *Bevins et al., 2011*; *Simpson and Goodhill, 2011*).

In the following step, the gradient of experimentally measured retinal Efnas is transposed to the SC according to the layout of the RC map (*Figure 2C,I,O*). For WT, this generates a smooth monotonically increasing gradient (*Figure 2C*). For the Isl2^Epha3/+ genotype, this transposition generates a double oscillatory gradient of retinal Efnas in the SC (*Figure 2I*), a consequence of the partial duplication of the RC map (*Figure 2K*). For the Isl2^Epha3/Epha3, the transposition of the retinal Efna gradients, according to the fully duplicated RC map (*Figure 2Q*), generated a double oscillatory gradient of transposed retinal Efnas in the SC (*Figure 2O*).

Finally, the projections of one hundred V1 neurons onto the SC (the CC map) were simulated for all three genotypes (*Figure 2F,L,R*). V1 axons/growth cones carrying gradients of Epha receptors (*Figure 2D,J,P*) are repelled by the transposed gradients of retinal Efna into the SC (forward signaling) as suggested previously (*Savier et al., 2017*). The WT CC map is smooth and continuous (*Figure 2F*), similarly to the RC map (*Figure 2E*). The Isl2^Epha3/+ simulations show dispersed projections forming two separated maps collapsing in the rostral-most part of the SC from lateral-most V1 neurons (*Figure 2L*). CC map simulations for the Isl2^Epha3/Epha3 genotype shows dispersed projections forming two fully separated maps (*Figure 2R*).

**Table 1.** summary of the parameters of the 3-step map alignment algorithm.

| Receptor | Epha3 | Epha4 | Epha5 | Epha6 | Source |
|---|---|---|---|---|---|
| Retina | WT = 0<br>Epha3^KI/KI = 1.86<br>Epha3^KI/+ = 0.93 | 1.05 | $0.14e^{0.018x}$ | $0.09e^{0.029x}$ | Measured (*Brown et al., 2000*; *Reber et al., 2004*) |
| V1 | $e^{(-x/100)} - e^{((x - 200/100) + 1)}$ | | | | Estimated (*Tsigankov and Koulakov, 2010*; *Tsigankov and Koulakov, 2006*) |
| Ligand | Efna2 | Efna3 | | Efna5 | |
| Retina | $1.85\,e^{-0.008x}$ | 0.44 | | $1.79\,e^{-0.014x}$ | Measured (*Savier et al., 2017*) |
| SC | $e^{((x - 100)100)} - e^{((-x-100)/100)}$ | | | | Estimated (*Cang et al., 2005*; *Savier et al., 2017*; *Tsigankov and Koulakov, 2010*; *Tsigankov and Koulakov, 2006*) |
| **Parameters** | | | | | |
| $\gamma$ | 1 | | | Strength of activity interaction | |
| $\alpha$ | 200 | | | Chemical strength | |
| d | 3 | | | SC interaction distance | |
| b | 0.11 | | | Retinal correlation distance | |

## Experimental validation of the 3-step map alignment model

To validate the simulations of the model predicting the formation of visual maps in the Isl2-Epha3KI animals, we performed anterograde RC and CC tracings in vivo. In the Isl2$^{Epha3/+}$ animals, two experimental measurements (*Figure 3A*, triangles and square in *Figure 3B* – *Figure 3—source data 1B*) are shown, plotted on the simulated RC map (black dots, gray/black lines, *Figure 3B*) indicating a duplicated projection for cells from the nasal pole of the retina as extensively shown earlier (*Brown et al., 2000*; *Reber et al., 2004*; *Savier et al., 2017*). For RGCs located on the temporal side of the retina, a single projection can be found as was also described previously (*Brown et al., 2000*; *Reber et al., 2004*; *Savier et al., 2017*). Our experimental measurements (triangles and square, *Figure 3B* – *Figure 3—source data 1B*) match with the theoretical representation (black dots, black/gray lines *Figure 3B*). Further validation of the algorithm was performed by in vivo analysis of the CC mapping in the Isl2$^{Epha3/+}$ animals. CC anterograde tracing (*Figure 3D*, n = 15, red dots/lines – *Figure 3—source data 1D*), performed as described earlier (*Savier et al., 2017*), revealed a partially duplicated CC map with the occurrence of a collapse point near 82% of the medial-lateral axis of V1 (16% of the rostral-caudal axis of the SC), similarly to the simulated map (black dots, black/gray lines *Figure 3D*, two-samples Kolmogorov-Smirnov test). Two of the injections and their corresponding terminations zones in the SC are depicted (*Figure 3C*, arrows and arrowhead in *Figure 3D*).

In Isl2$^{Epha3/Epha3}$, RC duplications are observed as shown by two experimental measurements (triangles and squares) plotted on the RC map (*Figure 3E,F* – *Figure 3—source data 1F*) and as extensively described previously (*Brown et al., 2000*; *Reber et al., 2004*). These experimental data match with the simulated mapping (black dots, black/gray lines *Figure 3F*). CC map tracing in vivo (*Figure 3G,H*, red dots/line, n = 7 – *Figure 3—source data 1H*) revealed a fully duplicated CC map, as predicted by the simulations (*Figure 3H*, black dots, black/gray lines, two-samples Kolmogorov-Smirnov test). Two of the injections and their corresponding terminations zones in the SC are depicted (*Figure 3G*, arrows and arrowhead in *Figure 3H*). These results further validate the predictions and reinforce the underlying mapping principle encoded in the 3-step map alignment algorithm.

## Retino/cortico-collicular maps organization indexes

Previous work demonstrated that RC and CC maps must be aligned and in register to allow efficient detection of visual stimuli by the SC (*Zhao et al., 2014*; *Liang et al., 2015*; *Basso and May, 2017*). We further implemented the 3-step map alignment algorithm to calculate an 'intrinsic dispersion index' (IDI) for each visual map (IDI$_{retino}$ and IDI$_{cortico}$) and an 'alignment index' (AI) for each genotype. These indexes measure the overall organization of the RC and CC maps. IDI$_{retino}$ and IDI$_{cortico}$ indicate the degree of dispersion of the corresponding map (or within-map variability) and should be minimum. AI represents the degree of alignment between the RC and CC maps and should be close to 1. AI = 1 corresponds to the theoretical one-to-one alignment of all the RGCs projections with all the V1 projections onto the SC, however such value will not be observed due to the intrinsic variability of the mapping introduced by the stochastic process of spontaneous activity. In WT animals, median IDI$_{retino}$ = 5.66, 95% CI [5.15; 6.17], median IDI$_{cortico}$ = 8.12 [7.63; 8.60] and median AI = 2.23 [2.13; 2.33] (*Figure 4A,G* – *Figure 4—source data 1A, G*). These values correspond to the control values for aligned, single RC and CC maps (*Figure 4B*, black and white dots). In Isl2$^{Epha3/+}$, median IDI$_{retino}$ = 38.8 [36.8; 40.8], median IDI$_{cortico}$ = 38.0 [36.1; 40.0] and median AI = 2.07 [1.95; 2.19] (*Figure 4A* – *Figure 4—source data 1A*) indicating increased spreading of both RC and CC projections, due to the partially duplicated maps (*Figure 4C*, black and white dots). The median AI value in Isl2$^{Epha3/+}$ (2.07 [1.95, 2.19]) is not statistically different from WT (*Figure 4G* – *Figure 4—source data 1G*), suggesting that Isl2$^{Epha3/+}$ RC/CC maps are aligned (*Figure 4C*, black and white dots). In the Isl2$^{Epha3/Epha3}$ animals, median IDI$_{retino}$ = 51.7 [50.8; 52.6], median IDI$_{cortico}$ = 50.6 [49.7; 51.5] and median AI = 2.2 [2.08; 2.32] (*Figure 4A,G* - *Figure 4—source data 1A,G*), indicating an increased separation between the RC and CC maps compared to WT and Isl2$^{Epha3/+}$ due to fully duplicated projections (*Figure 4D* white and black dots). The median AI value (2.20 [2.02; 2.43]) is not significantly different from WT (*Figure 4G* – *Figure 4—source data 1G*), indicating aligned RC and CC maps (*Figure 4D*, black and white dots). Median AI values for WT, Isl2$^{Epha3/+}$ and Isl2$^{Epha3/}$

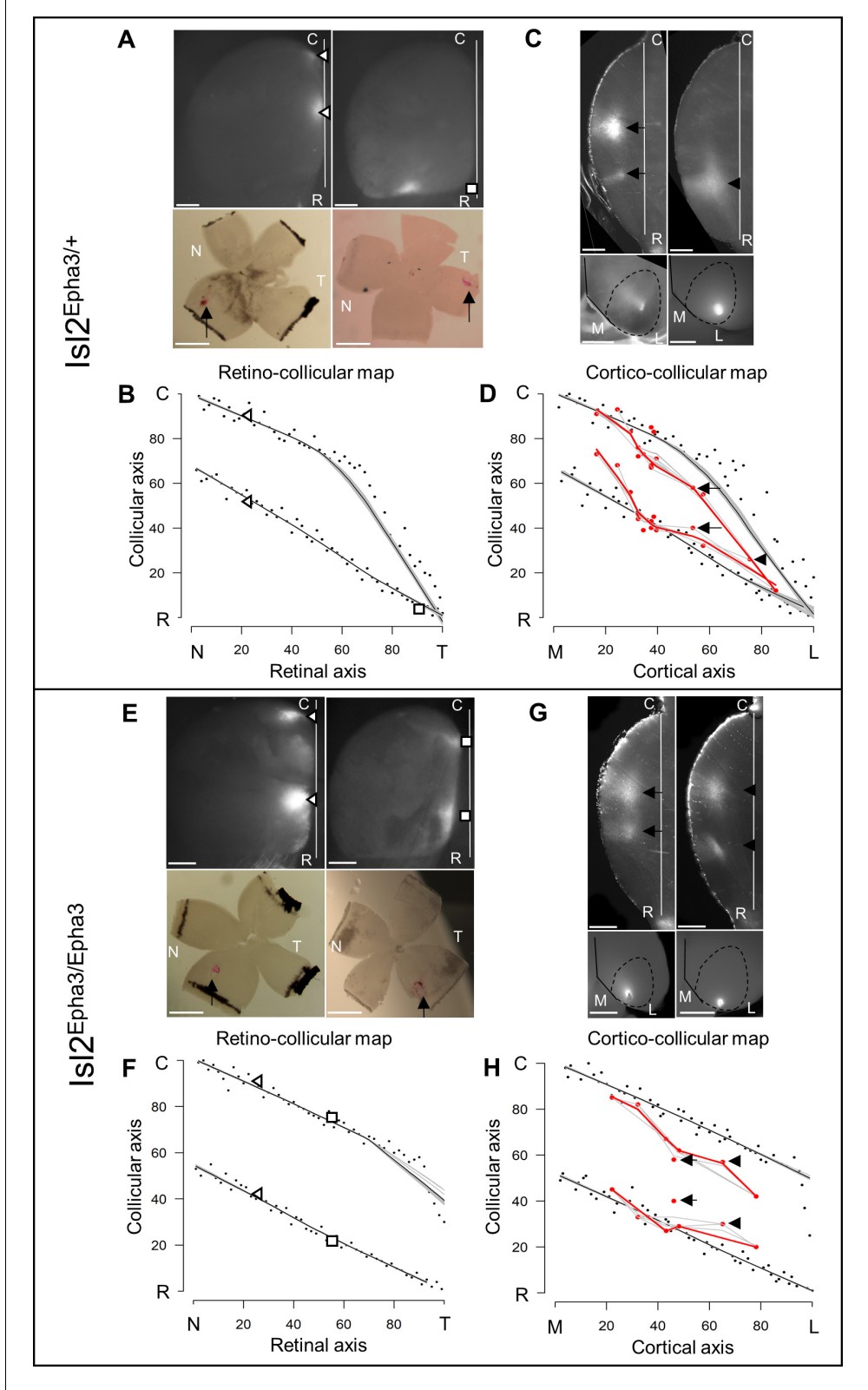

**Figure 3.** Experimental validation of retino- and cortico-collicular mapping in Isl2-Epha3KI animals. (**A**) Images of two experimental injections showing the collicular terminations zones (triangles and square, top-view, upper panels) after focal retinal injections (arrows, flat-mount, lower panel) in Isl2$^{Epha3/+}$ animals. (**B**) Cartesian representation of the injections (triangles and square) in (**A**) superimposed with the simulated RC map (black dots,

*Figure 3 continued*

n = 100) in Isl2$^{Epha3/+}$. Map profile is calculated by LOESS smoothing (black and gray lines). (C) Images of two experimental injections showing the collicular termination zones (sagittal view, upper panels) after focal cortical V1 injection (top-view, lower panels). Arrows and arrowheads indicate the site of the termination zones. Lower left panel shows CO staining (dark gray) delineating V1. (D) Cartesian representation of the experimental (red dots/ lines, n = 15 animals) and simulated (black dots, n = 100) CC maps calculated by LOESS smoothing (black, red and gray lines). Arrows and arrowhead represent the two injections shown in (C). Two-samples Kolmogorov-Smirnov test, D-stat = 0.273 < D-crit.=0.282, p=0.06, simulated and experimentally measured CC maps are not significantly different. (E) Images of two experimental injections showing the collicular terminations zones (triangles and squares, top-view, upper panels) after focal retinal injection (arrows, flat-mount, lower panel) in Isl2$^{Epha3/Epha3}$ animals. (F) Cartesian representation of the injections (triangles and squares) in (E) superimposed with the simulated RC map (black dots, n = 100) in Isl2$^{Epha3/Epha3}$. Map profile is calculated by LOESS smoothing (black and gray lines). (G) Images of two experimental injections showing the collicular duplicated termination zones (arrows and arrowheads, sagittal view, upper panels) after focal cortical V1 injection (top-view, lower panels). (H) Cartesian representation of the experimental (red dots/lines, n = 7 animals) and simulated (black dots, n = 100) CC maps calculated by LOESS smoothing (black, red and gray lines). Arrows and arrowheads represent the two examples in (G). Two-samples Kolmogorov-Smirnov test, D-stat = 0.190 < D-crit.=0.371, p=0.72, simulated and experimentally measured CC maps are not significantly different. Scale bars: 400 µm (A upper, C, E upper, G), 1 mm (A, E lower). Abbreviations: N, nasal; T, temporal; R, rostral; C, caudal; M, medial; L, lateral.

The online version of this article includes the following source data for figure 3:

**Source data 1.** Data for *Figure 3B*: Cartesian values of experimentally measured collicular termination zones and corresponding retinal injections in Isl2$^{Epha3/+}$ animals.

---

$^{Epha3}$ are not significantly different suggesting that RC and CC maps in these animals are aligned (*Figure 4G* – *Figure 4—source data 1G*), although partially or fully duplicated.

To further test and validate these map organization indicators, we calculated the IDIs and AI in previously characterized Isl2$^{Efna3/+}$ and Isl2$^{Efna3/Efna3}$ animals (*Savier et al., 2017*). Median values for Isl2$^{Efna3/+}$ are IDI$_{retino}$ = 5.56 [5.04; 6.07] and IDI$_{cortico}$ = 12.5 [11.8; 13.2] (*Figure 4A* – *Figure 4— source data 1A*). IDI$_{retino}$ is not significantly different from WT suggesting no dispersion of the RC map (*Figure 4E*, white dots). However, IDI$_{cortico}$ is significantly different from WT indicating a spreading of the CC map (*Figure 4E*, black dots). This suggests a misalignment between RC and CC maps (*Figure 4E*) as confirmed by the median AI value in Isl2$^{Efna3/+}$ (*Figure 4G*, 4.52 [4.28, 4.76] - *Figure 4—source data 1G*), significantly different from WT. Visual map misalignment is more pronounced in Isl2$^{Efna3/Efna3}$ (median AI = 8.85 [8.23, 9.26], *Figure 4G* – *Figure 4—source data 1G*) compared to Isl2$^{Efna3/+}$ as previously shown in vivo (*Savier et al., 2017*). Map dispersion value IDI$_{retino}$ in Isl2$^{Efna3/Efna3}$ indicates that RC map spreading is similar to WT (*Figure 4F*, white dots; *Figure 4A*, median IDI$_{retino}$ = 5.56 [5.08, 6.03], not significantly different from WT) whereas IDI$_{cortico}$ indicates dispersion of the CC map (*Figure 4F*, black dots; *Figure 4A*, median IDI$_{cortico}$ = 19.1 [17.7, 20.5] – *Figure 4—source data 1A*) as demonstrated previously (*Savier et al., 2017*). Such significant dispersion of the CC map leads to an important misalignment between RC and CC projections in Isl2$^{Efna3/Efna3}$ animals (median AI = 8.85 [8.59; 9.11] significantly different from WT, *Figure 4F,G*).

We undertook a more detailed analysis of the mapping organization and alignment by implementing the local 'intrinsic dispersion variation' (IDV) (y axis) for each RC and CC maps along the rostral-caudal axis of the SC (x axis, *Figure 5—source data 1*). Median local IDVs indicate the degree of dispersion and alignment as a function of the position along the rostral-caudal axis in the SC. In WT (*Figure 5A*), Isl2$^{Epha3/+}$ (*Figure 5B*) and Isl2$^{Epha3/Epha3}$ (*Figure 5C*), median local IDVs between RC and CC maps are similar and covary (> 0) (WT RC/CC: Jaccard similarity index = 0.60 indicating 60% overlap- covariance = 1.64; Isl2$^{Epha3/+}$RC/CC: Jaccard similarity index = 0.30, covariance = 88.2; Isl2$^{Epha3/Epha3}$ RC/CC: Jaccard similarity index = 0.55, covariance = 12.8), indicating that the maps are aligned. This is in sharp contrast to the Isl2$^{Efna3/+}$ (*Figure 5D*) and Isl2$^{Efna3/Efna3}$ (*Figure 5E*) animals where median local RC and CC IDVs variation do not superimpose nor covary (Isl2$^{Efna3/+}$RC/CC: Jaccard similarity index = 0.17, covariance = −1.99; Isl2$^{Efna3/Efna3}$RC/CC: Jaccard similarity index = 0.08, covariance = −4.13). Moreover, values of IDV are indicative of map dispersion along the RC axis in the SC. We calculated a cut-off value of median local IDV based on WT RC/CC maps organization (median local IDV$_{threshold}$ = 14.7, the minimum local IDV, dashed black line in *Figure 5*) corresponding to the WT maps dispersion. Values greater than the local IDV$_{threshold}$, indicate a

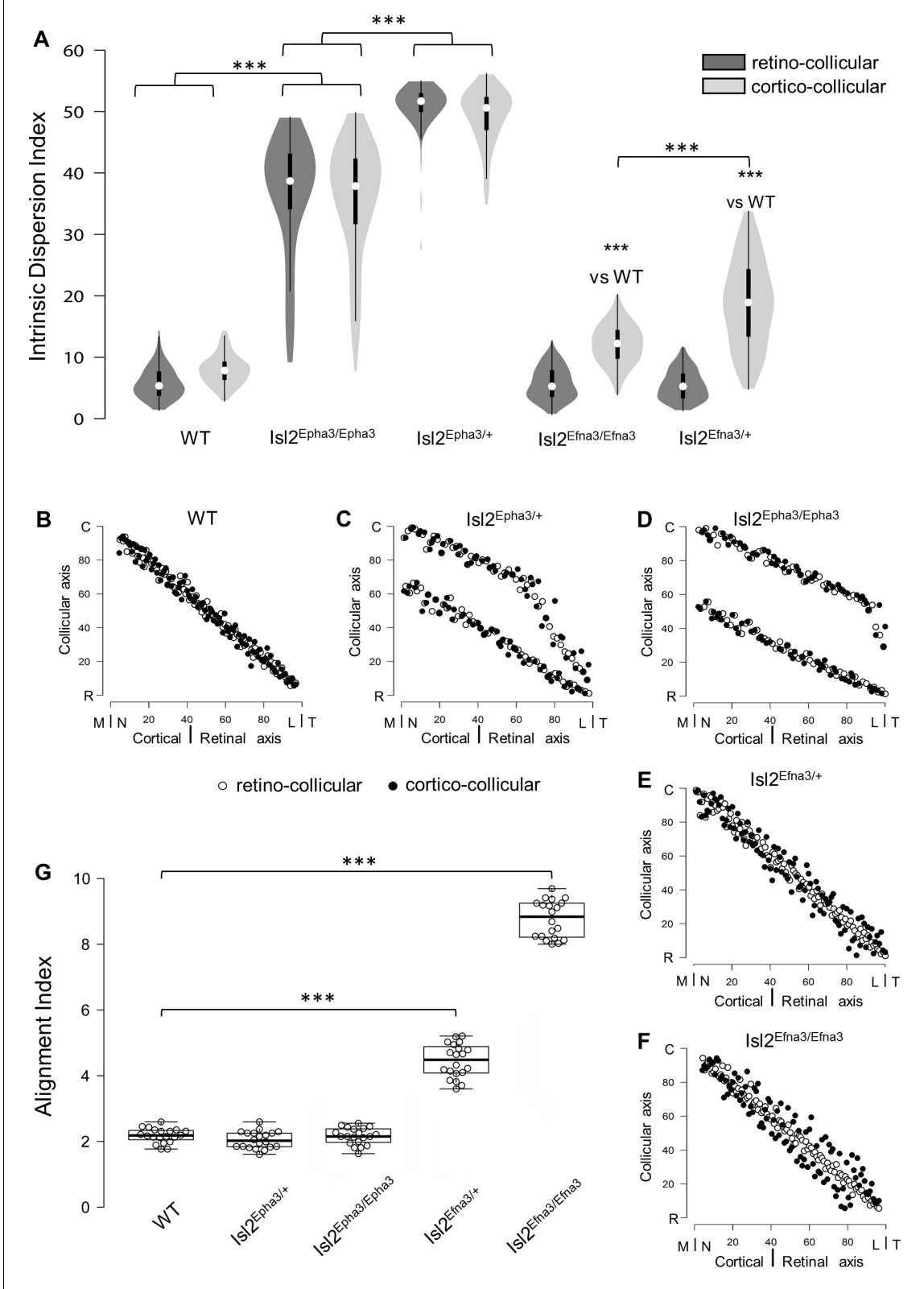

**Figure 4.** Intrinsic dispersion index (IDI) and alignment index (AI) in Isl2-Epha3KI and Isl2-Efna3KI animal models. (**A**) Violin plot representation of the median IDIs (from n = 10 simulated maps, each composed of 100 projections) in WT, Isl2-Epha3KI and Isl2-Efna3KI animal models. Mann-Whitney test: WT IDI$_{retino}$ vs. Isl2$^{Epha3/+}$ IDI$_{retino}$, z-score = 12.08, effect r = 0.85, p=6E-55; WT IDI$_{cortico}$ vs. Isl2$^{Epha3/+}$ IDI$_{cortico}$, z-score = 12.04, effect r = 0.85, p=8E-52; Isl2$^{Epha3/+}$ IDI$_{retino}$ vs Isl2$^{Epha3/Epha3}$ IDI$_{retino}$, z-score = 11.25, effect r = 0.80, p=2E-39; Isl2$^{Epha3/+}$ IDI$_{cortico}$ vs Isl2$^{Epha3/Epha3}$ IDI$_{cortico}$, z-score = 10.53, effect

*Figure 4 continued on next page*

Figure 4 continued

r = 0.74, p=1E-32; WT $IDI_{cortico}$ vs $Isl2^{Efna3/+}$ $IDI_{cortico}$, z-score = 8.30, effect r = 0.59, p=1E-18; WT $IDI_{cortico}$ vs $Isl2^{Efna3/Efna3}$ $IDI_{cortico}$, z-score = 10.37, effect r = 0.73, p=3E-31; $Isl2^{Efna3/+}$ $IDI_{cortico}$ vs $Isl2^{Efna3/Efna3}$ $IDI_{cortico}$, z-score = 6.93, effect r = 0.49, p=6E-13; ***p<0.001. (B, C, D, E, F) Representation and superimposition of simulated RC (retino-collicular) (white dots) and CC (cortico-collicular) (black dots) maps in WT (B), $Isl2^{Epha3/+}$ (C), $Isl2^{Epha3/Epha3}$ (D), $Isl2^{Efna3/+}$ (E) and $Isl2^{Efna3/Efna3}$ (F) animals (representative of n = 10 runs). (G) Box plot representation of median AI (from n = 20 simulated RC/CC maps) in WT, $Isl2^{Epha3/+}$, $Isl2^{Epha3/Epha3}$, $Isl2^{Efna3/+}$ and $Isl2^{Efna3/Efna3}$ animals. Mann-Whitney test: AI WT vs. AI $Isl2^{Epha3/+}$, z-score = 1.62, effect r = 0.26 p=0.10; AI WT vs. AI $Isl2^{Epha3/Epha3}$, z-score = 0.11, effect r = 0.02, p=0.90; AI WT vs AI $Isl2^{Efna3/+}$, z-score = 5.40, effect r = 0.85, p=1.45E-11; AI WT vs AI $Isl2^{Efna3/Efna3}$, z-score = 5.40, effect r = 0.85, p=1.45E-11. ***p<0.001. Abbreviations: IDI, intrinsic dispersion index; AI, alignment index; WT, wild-type; N, nasal; T, temporal; R, rostral; C, caudal; M, medial; L, lateral.

The online version of this article includes the following source data for figure 4:

**Source data 1.** Data for *Figure 4A*: Intrinsic Dispersion Index values for retino-collicular (RC) and cortical-collicular (CC) mapping in wild-type (WT), Isl2-Epha3KI and Isl2-Efna3KI animals.

duplication of the map, while values below this threshold indicate a single map. In WT (*Figure 5A*, yellow), $Isl2^{Efna3/+}$ (*Figure 5D*, light red) and $Isl2^{Efna3/Efna3}$ (*Figure 5E*, light blue), median local $IDV_{retino}$ representations suggest a similar dispersion along the RC axis with all IDV values below $IDV_{threshold}$ indicating single maps and corresponding to experimental observations (*Savier et al., 2017*). In $Isl2^{Epha3/+}$ $IDV_{retino}$ and $IDV_{cortico}$ (*Figure 5B*), $Isl2^{Epha3/Epha3}$ $IDV_{retino}$ and $IDV_{cortico}$ (*Figure 5C*), $Isl2^{Efna3/+}$ $IDV_{cortico}$ (*Figure 5D*, dark red) and $Isl2^{Efna3/Efna3}$ $IDV_{cortico}$ (*Figure 5E*, dark blue), values are above $IDV_{threshold}$, indicating map duplication. More precisely, in $Isl2^{Epha3/+}$ animals, median local $IDV_{retino}$ and median local $IDV_{cortico}$ reach $IDV_{threshold}$ at 12% and at 8% of the rostral-caudal axis in the SC respectively (*Figure 5B*) consistent with the position of the collapse points described earlier (*Brown et al., 2000*; *Savier et al., 2017* and *Figure 3B,D*). This suggests single RC and CC maps in the rostral-most pole of the SC and a duplicated visual map from 8–12% to 100% of the rostral-caudal axis as shown experimentally and theoretically (*Brown et al., 2000*; *Savier et al., 2017*; *Figure 3B,D*). In $Isl2^{Epha3/Epha3}$ animals, both RC and CC median local IDV > $IDV_{threshold}$ along the entire rostral-caudal axis (*Figure 5C*), indicating full RC and CC maps duplication as shown experimentally (*Brown et al., 2000*; *Triplett et al., 2009*; *Figure 3F,H*). Finally, for both $Isl2^{Efna3/+}$ and $Isl2^{Efna3/Efna3}$ CC maps (*Figure 5D*, dark red, *Figure 5E*, dark blue, respectively) local IDVs oscillate around the $IDV_{threshold}$ with a more pronounced effect in the rostral-half of the SC in $Isl2^{Efna3/Efna3}$ indicating partial duplications of the CC maps in these animals, as demonstrated in vivo (*Savier et al., 2017*). RC median local IDVs in $Isl2^{Efna3/+}$ and $Isl2^{Efna3/Efna3}$ animals (*Figure 5D* pink line, *Figure 5E* light blue line, respectively) fall below $IDV_{threshold}$ indicating single maps as demonstrated previously (*Savier et al., 2017*). Altogether these results suggest that IDIs, AIs and median local IDVs are relevant, reliable and accurate indicators of visual map organization and conformation.

## Discussion

### The 3-step map alignment model replicates other visual map-defective mutants

Taken together, these experimental and theoretical results confirm the validity and robustness of the 3-step map alignment algorithm in predicting RC and CC maps formation and alignment during development. These findings broaden the simulation capacity of the model to the formation and alignment of visual maps in Isl2-Epha3KI mutant animals originally described in 2000 (*Brown et al., 2000*). After $10^7$ iterations per run, the algorithm generates RC maps for $Isl2^{Epha3/+}$ heterozygous and $Isl2^{Epha3/Epha3}$ homozygous animals. In $Isl2^{Epha3/Epha3}$, the simulated RC map is fully duplicated, as described in vivo (*Brown et al., 2000*; *Reber et al., 2004*; *Figures 2Q* and *3F*) whereas in the $Isl2^{Epha3/+}$, the simulated RC duplication collapses at approximately 80% of the nasal-temporal axis, similarly to previous results (*Brown et al., 2000*; *Reber et al., 2004*; *Figures 2K* and *3B*). More recent experimental analyses show a collapse point appearing within a range of 74% to 80% of the nasal-temporal axis (*Savier et al., 2017*) in contrast to previous results indicating a collapse at 76% of the nasal-temporal axis (*Brown et al., 2000*; *Reber et al., 2004*). This discrepancy might be explained by the measurement methods which differ between the original *Brown et al., 2000* publication and the recent *Savier et al., 2017* and here (*Figure 3B,D,E,F*). In the last two, the Locally

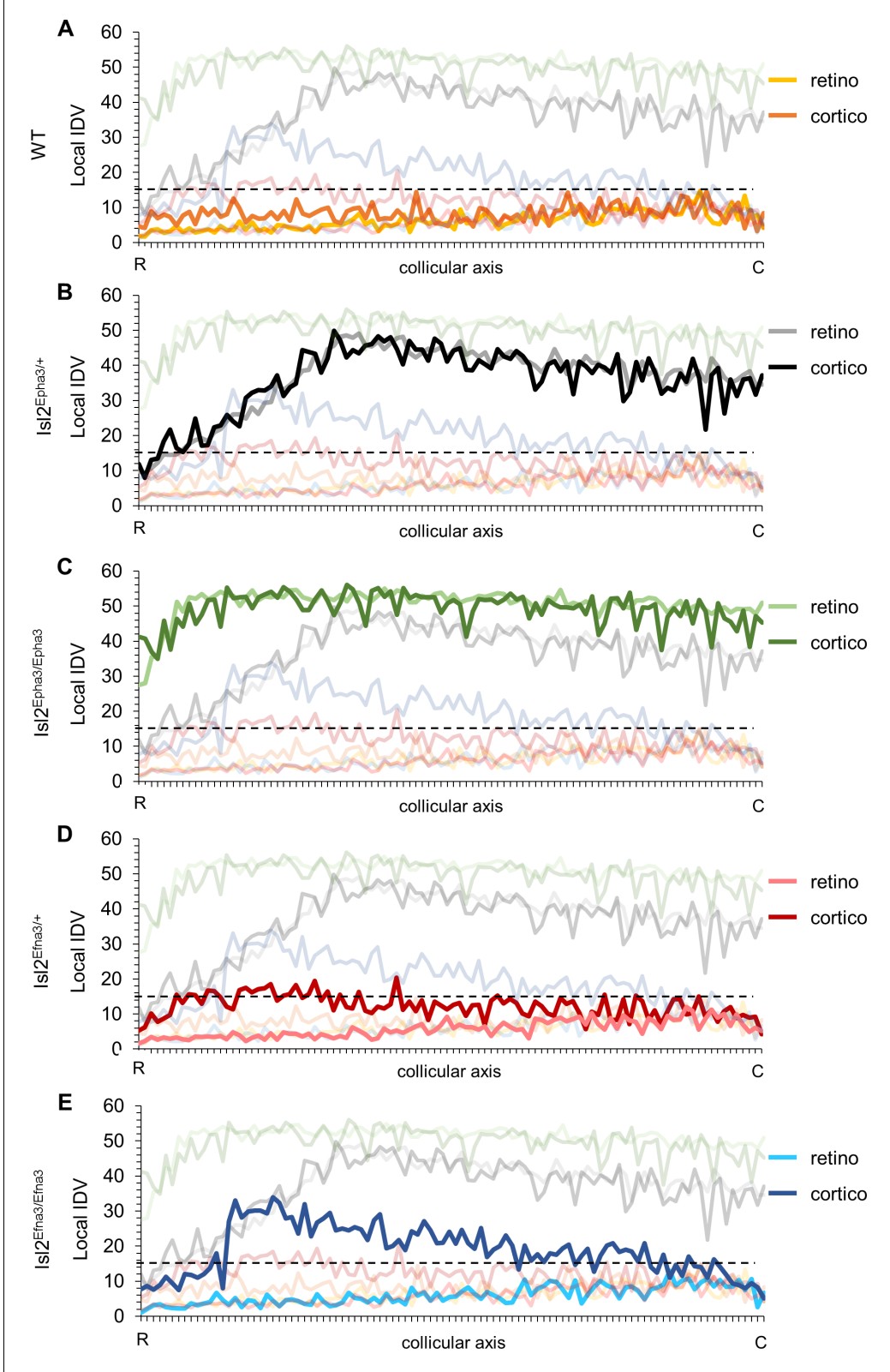

**Figure 5.** Local intrinsic dispersion variation (local IDV) in WT, Isl2-Epha3KI and Isl2-Efna3KI animal models. (A, B, C, D, E) Representation of the local IDV values for both retinal (light colors) and cortical (dark colors) projections along the rostral R (0%) – caudal C (100%) axis of the SC in WT (A), Isl2$^{Epha3/+}$ Isl2$^{Epha3/+}$ (B), Isl2$^{Epha3/Epha3}$ (C), Isl2$^{Efna3/+}$ (D) and Isl2$^{Efna3/Efna3}$ (E) animals (representative of n = 10 runs). Dashed line represents the threshold above which maps are duplicated. Abbreviations: IDV, intrinsic dispersion variation; WT, wild-type; R, rostral; C, caudal.

*Figure 5 continued on next page*

*Figure 5 continued*

The online version of this article includes the following source data for figure 5:

**Source data 1.** Local intrinsic dispersion variation (local IDV) values for retino-collicular (RC) and cortical-collicular (CC) mapping in wild-type (WT), Isl2-Epha3KI and Isl2-Efna3KI animals.

Weighted Scatterplot Smoothing (LOWESS) cross validation method (*Efron and Tibshirani, 1991*) was used on both experimental and simulated maps providing a range of values, instead of a given value as performed in *Brown et al., 2000*, for the occurrence of the collapse point along the nasal-temporal axis (see also Figure 6—figure supplement 1 in *Savier et al., 2017*). In Isl2-Epha3KI animals, the lower RC map corresponds to Isl2(+) Epha3-expressing RGCs, covering 0 to 50% of the rostral-caudal axis in the Isl2$^{Epha3/Epha3}$ and 0 to 80% of the axis in Isl2$^{Epha3/+}$ (Figures 5 and 6 in *Brown et al., 2000*). The upper map, corresponding to the Isl2(-) WT RGCs, covers the caudal half (50 to 100%) of the rostral-caudal axis of the SC in Isl2$^{Epha3/Epha3}$ and 20–100% of the axis in Isl2$^{Epha3/+}$ (Figures 5 and 6 in *Brown et al., 2000*). In both contexts, nasal Isl2(-) RGCs, expressing high levels of Efnas, project to caudal locations in the SC whereas nasal Isl2(+) RGCs, also carrying high levels of Efnas, project ectopically in a more rostral part of the SC, where the WT levels of retinal Efnas are normally low (compare *Figure 2C* with *Figure 2I and O*; see also *Figure 6*). Such distribution of RGC projections in the SC in both Isl2$^{Epha3/Epha3}$ and Isl2$^{Epha3/+}$ animals leads to a perturbation of the transposed retinal Efna gradients in the SC, generating duplicated Efna gradients along the rostral-caudal axis in both genotypes (*Figure 2I,O*, *Figure 6*). Consequently, simulated CC maps are also duplicated for both Isl2$^{Epha3/+}$ and Isl2$^{Epha3/Epha3}$ and further align with the duplicated RC maps as confirmed by in vivo analyses (*Figure 3D,H*, *Figure 6*). Further validation of the 3-step alignment model would involve testing this algorithm using other genetic models such as the Math5KO, which presents an unrefined CC map. The prediction of the CC map phenotype of compound mutants such as Efnas triple knock-out and Epha4KO crossed with Isl2-Epha3KI would also be of prime interest but would require the experimental assessment of the CC map in these mutants, which to our knowledge, has not been performed to this date.

## The basic map organization principle encoded in the 3-step map alignment model provides an alternative explanation for map alignment defects in Isl2-Epha3KI animals

Our quantification of Ephas and Efnas throughout the visual system revealed that the ectopic over-expression of Epha3 in the retina has no effect on the expression of Efnas in this structure. These findings exclude a potential alteration of the endogenous gradient of Efnas in the retina. Previous studies (*Triplett et al., 2009*) suggested that the redistribution of the correlated activity was responsible for the duplication of the CC map. Our results suggest an alternative, although complementary mechanism, in which the redistribution of the molecular cues carried by retinal axons during the formation of a duplicated RC map is sufficient to induce a collicular duplication of the projections coming from V1 (*Figure 6*). As mentioned earlier (*Savier et al., 2017*), in this context retinal Efnas, free or bound to Ephas to a limited amount (*Suetterlin and Drescher, 2014*), retain their binding activity in the SC for incoming V1 axons carrying Epha receptors. Since the RC map is duplicated in the Isl2-Epha3KI, the nasal-temporal axis of the retina, along which the retinal Efna gradients run, is represented twice in the collicular space. This rearrangement of the retinal Efnas transposed to the SC generates a duplicated gradient which is then encountered by the incoming cortical V1 axons. At this point in development, the endogenous collicular Efnas are not available anymore as, for the most part, they have been endocytosed upon trans-binding with Epha receptors on retinal axons (*Janes et al., 2005*; *Savier et al., 2017*). As a consequence, the optimal local amount of transposed Efnas signaling the corresponding retinotopic position to V1 axons exists at two distinct locations along the rostral-caudal axis, leading to duplication of the CC map. Our previous findings in the characterization of the Isl2-Efna3KI mutant revealed an incomplete penetrance of the CC duplication phenotype, suggesting a counter-balancing role of the correlated activity within the normal RC map. For the Isl2-Epha3KI, the correlated activity is also duplicated along the rostro-caudal axis, due to the duplicated RC map, reinforcing this 'duplication' effect and leading to a stronger penetrance of the phenotype. To further test this hypothesis, it would be interesting to selectively manipulate

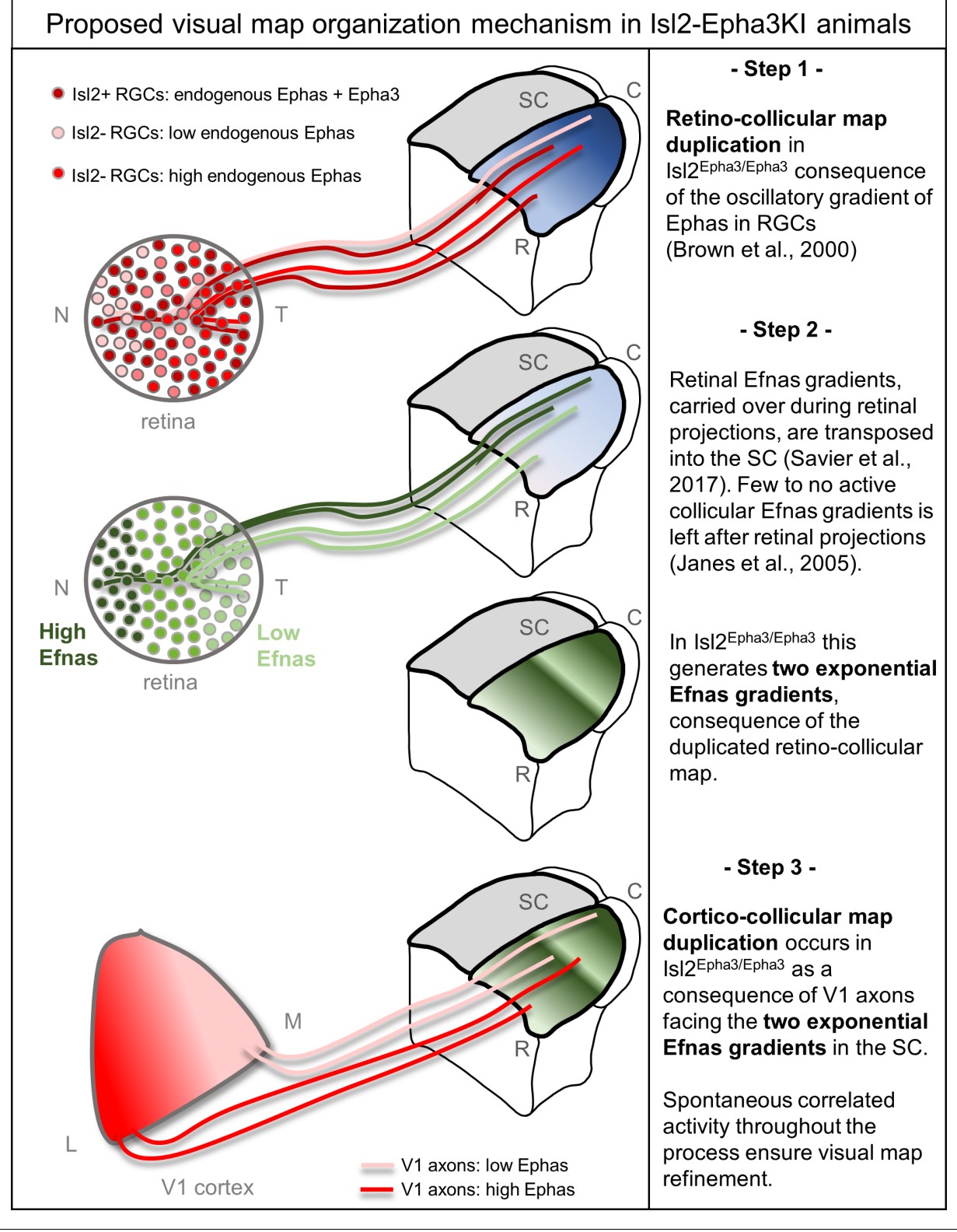

**Figure 6.** Proposed mechanism of visual map duplication and alignment in Isl2-Epha3KI animals based on the 3-step map alignment model. Step 1: Isl2 (+) RGCs expressing WT levels Ephas + ectopic Epha3 (dark red) and Isl2(-) RGCs expressing WT levels of Ephas (light red to red) send their axons into the SC during the first postnatal week. These retino-collicular (RC) projections form a duplicated map due to the oscillatory gradient of Ephas receptors in the RGCs reading the WT collicular Efna gradients (blue R-C gradient in SC) through forward signaling. Step 2: Retinal Efna gradients (high Efnas-

*Figure 6 continued on next page*

**Figure 6 continued**

nasal-dark green, low Efnas-temporal-light green) are carried to the SC during the formation of the RC projections. This transposition of retinal Efnas generates two exponential gradients of Efna in the SC, due to the duplication of the RC map and replaces the WT collicular Efna gradients previously used by the RGCs axons (*Janes et al., 2005*). Step 3: V1 axons, expressing smooth gradients of Ephas receptors (light red – red), are facing two exponential gradients of Efnas, of retinal origin, in SC. Through forward signaling, this two exponential Efna gradients generate a duplication of the CC projections, which aligns with the RC map. Abbreviations: N, nasal; T, temporal; M, medial; L, lateral; R, rostral; C, caudal; SC, superior colliculus; RGCs, retinal ganglion cells.

correlated activity during development in either the retina, the SC or V1. Our current conceptual framework suggests that altering the activity in the SC would still result in a duplication of the CC map of the Isl2-Epha3KI and an increase in the amount of CC duplication in the Isl2-Efna3KI. The 3-step map alignment model suggests that retinal Efnas are responsible for the CC map duplication in the Isl2-Epha3KI mutants. An additional validation of this mechanism would require the mapping of Isl2-Epha3KI CC map in compound mutants in which Efnas have been specifically removed in the retina (*Suetterlin and Drescher, 2014*). The absence of duplication in the CC map in such a mutant would further demonstrate the implication of retinal Efnas in the alignment of visual maps in the SC. The mapping and alignment along the lateral-medial axis of the SC involving EphB/Efnb signaling is not addressed here as the Isl2-Epha3KI animals only present mapping defect along the rostral-caudal axis (*Brown et al., 2000*; *Triplett et al., 2009*).

## Quantifying the degree of maps alignment and organization

We performed a quantitative analysis of the degree of visual map organization, implementing the intrinsic dispersion index (IDI) for each visual map, the mean alignment index (AI) (*Figure 4—source data 1*) and the local intrinsic dispersion variation (local IDV) (*Figure 5—source data 1*). These indexes objectively describe and quantify the degree of organization and conformation of both RC and CC maps, providing a detailed measure of the dispersion of each map and a measure of the degree of alignment between the maps as a whole and locally. In general, the experimental number of measured projections is low, approximately n = 15 to 20 total, compared to the n = 100 (or more) projections generated by the algorithm. This suggests that, if the algorithm accurately simulates maps organization, the indicators, IDI, AI, and local IDVs, inferred from this algorithm are more precise and refined than those inferred from experimental maps. As demonstrated above, the 3-step map alignment algorithm simulates and predicts visual map defects in Isl2-Epha3KI animals as well as in other animal models (*Savier et al., 2017*) indicating its reliability and robustness; therefore, we inferred the IDIs, AIs and local IDVs from the algorithm-generated maps for each genotype. An increase of both IDIs and local IDVs in $\text{Isl2}^{Epha3/+}$ and $\text{Isl2}^{Epha3/Epha3}$ animals correlates with increased spreading of the visual maps, consequence of the duplicated RC and CC projections. By contrast, AI values in these animals are similar to WT, indicating alignment of the RC and CC maps, as observed in vivo. The relevance of the 3-step map alignment algorithm and the validity of the map organization indexes are further confirmed by the analysis of map organization in the previously characterized Isl2-Efna3KI animals. Here, IDIs, AIs and local IDVs generated by the algorithm predict visual map organization as previously shown in vivo (*Savier et al., 2017*). High AI values (>2.5 = WT median + 95% CI) indicate a significant misalignment between RC and CC projections, in particular in $\text{Isl2}^{Efna3/Efna3}$ (confirmed by the low Jaccard similarity index and negative covariance). Such misalignment is also indicated by a significant difference in dispersion (IDI) of the CC map when compared to the RC map ($\text{IDI}_{retino} \neq \text{IDI}_{cortico}$). These observations are further validated and confirmed by the local IDVs graph (*Figure 5—source data 1*) which delivers detailed information about the conformation of the maps along the rostral-caudal axis of the SC. Descriptive analyses of the IDVs using Jaccard similarity index and covariance guaranties robust data as to the degree of similarity of the maps (*Metcalf and Casey, 2016*). The implementation of these indexes provides a very detailed and robust analysis of the layout and a qualitative measure of the organization and alignment of the visual maps in different mice genetic models. It would also be interesting to characterize other compound mutants which present more separated maps but high alignment like the Isl2-Epha3KI/Epha4 KO (*Reber et al., 2004*) or the Isl2-Epha3KI/Epha5 KO (*Bevins et al., 2011*) double mutants.

# Materials and methods

## Key resources table

| Reagent type (species) or resource | Designation | Source or reference | Identifiers | Additional information |
|---|---|---|---|---|
| Genetic reagent (*Mus. musculus*) | Isl2<sup>tm1(Epha3)Grl</sup> | Lemke Lab (Salk Institute) | MGI:3056440 RRID:MGI:3057124 | |
| Commercial assay, kit | Neural tissue dissociation Kit – Postnatal neurons | Miltenyi | Cat # 130-094-802 | |
| Commercial assay, kit | RNeasy Mini kit | Qiagen | Cat # 74104 | |
| Commercial assay, kit | QuantiTect SYBR Green RT-PCR Kit | Qiagen | Cat # 204243 | |
| Antibody | Goat anti-rabbit IgG (H+L), polyclonal | Jackson Immunoresearch | Cat # 111-005-003 RRID:AB_2337913 | 1:3000 |
| Antibody | Goat anti-mouse IgM (H+L) | Biotrend | Cat # 610–1107 | 1:3000 |
| Antibody | Mouse anti-CD90, monoclonal | Biorad | Cat # MCA02R F7D5 IgM RRID:AB_323481 | 1:3000 |
| Antibody | Rabbit anti-rat macrophage, polyclonal | Life Science | Cat # AIA51240 | 1:3000 |
| Chemical compound, drug | Neurobasal medium | Gibco/Thermofisher | Cat # 21103049 | |
| Chemical compound, drug | B27 | Thermofisher | Cat # 17504044 | |
| Chemical compound, drug | BDNF | PeproTech | Cat # 450–02 | 25 ng/ml |
| Chemical compound, drug | CNTF | PeproTech | Cat # 450–13 | 10 ng/ml |
| Chemical compound, drug | Forskolin | Sigma | Cat # F3917 | 10 mM |
| Chemical compound, drug | Glutamine | Thermofisher | Cat # 25030149 | 2 mM |
| Chemical compound, drug | N-acetyl-l-cysteine | Sigma | Cat # A09165 | 60 mg/ml |
| Chemical compound, drug | Penicilin/streptomycin | Gibco | Cat # 15070–022 | 100 units/ml |
| Chemical compound, drug | Sodium pyruvate | Thermofisher | Cat # 11360070 | 1 mM |
| Chemical compound, drug | DiI (1,1-dioctadecyl-3,3,3,3-tetramethylindocarbocyanine perchlorate) | Thermofisher | Cat # D282 | |
| Chemical compound, drug | DiD (1,1'–dioctadecyl-3,3,3',3'-tetramethylindodicarbocyanine, 4-chlorobenzenesulfonate) | Thermofisher | Cat # D7757 | |
| Software, algorithm | R Project for Statistical Computing | www.r-project.org | RRID:SCR_001905 | |
| Software, algorithm | ImageJ | https://imagej.nih.gov/ij | RRID:SCR_003070 | |
| Software, algorithm | MATLAB | | RRID:SCR_001622 | |
| Software, algorithm | 3-step map alignment | Reber Lab (Krembil) | https://github.com/michaelreber/3-step-Map-Aligment-Model/blob/master/threestepsMapAlignment.m | |
| Software, algorithm | LOESS smoothing | Reber Lab (Krembil) | https://github.com/michaelreber/Leave-one-out-LOESS/blob/master/wtloess.R | |

### Retinal ganglion cell isolation

P1/P2 retinas were freshly dissected and RGCs were isolated and purified (>99%). For details, see *Steinmetz et al., 2006* and *Claudepierre et al., 2008*. Briefly, cells were harvested in neurobasal medium (Gibco/Invitrogen) supplemented with (all from Sigma, except where indicated) pyruvate (1 mM), glutamine (2 mM; Gibco/Invitrogen), N-acetyl-l-cysteine (60 mg/ml), putrescine (16 mg/ml), selenite (40 ng/ml), bovine serum albumin (100 mg/ml; fraction V, crystalline grade), streptomycin (100 mg/ml), penicillin (100 U/ml), triiodothyronine (40 ng/ml), holotransferrin (100 mg/ml), insulin (5 mg/ml) and progesterone (62 ng/ml), B27 (1:50, Gibco/Invitrogen), brain-derived neurotrophic factor (BDNF; 25 ng/ml; PeproTech, London, UK), ciliary neurotrophic factor (CNTF; 10 ng/ml; PeproTech) and forskolin (10 mM; Sigma). After isolation, RGCs were treated for RNA extraction.

### Quantitative RT-PCR

V1 cortices, superficial layers of the SC and retinas were freshly dissected. Retinas were cut in three equal pieces along the NT axis (Nasal, Central, Temporal RGCs) and RGCs acutely isolated (*Steinmetz et al., 2006*; *Claudepierre et al., 2008*; see above). Total RNA was extracted and quantified as previously described (*Savier et al., 2017*). Briefly, relative quantification was performed using the comparative Delta Ct method. Duplicates were run for each sample and concentrations for the target gene and for two housekeeping genes (hypoxanthine-guanine phosphoribosyl transferase - Hprt and glyceraldehyde 3-phosphate dehydrogenase – Gapdh) were computed.

### Projections analysis/mapping

Anterograde DiI (1,1-dioctadecyl-3,3,3,3-tetramethylindocarbocyanine perchlorate) or DiD (1,1'–dioctadecyl-3,3,3',3'-tetramethylindodicarbocyanine, 4-chlorobenzenesulfonate) labelling were performed blind to genotype as described (*Savier et al., 2017*). Retinas were dissected and imaged using Leica macroscope (MG0295) and LASAF software. RC projection coordinates of the DiI injections were calculated as described (*Brown et al., 2000*; *Reber et al., 2004*; *Bevins et al., 2011*). For CC map analyses, sagittal vibratome sections were performed on P14 SC and terminal zones (TZs) were plotted along the rostral-caudal axis on Cartesian coordinates (y axis). V1 were photographed as whole-mount and focal injections plotted along the V1 lateral-medial axis (x axis) (*Savier et al., 2017*; *Triplett et al., 2009*). CC maps were generated using non-parametric smoothing technique, termed LOESS smoothing (*Efron and Tibshirani, 1991*), to estimate the profile of the one-dimensional mapping from V1 to SC. To estimate the variability in a mapping containing N data points, we repeated the procedure N times with N-1 datapoints, each time dropping a different datapoint. This is termed a 'leave-one-out' method and was used in the R Project for Statistical Computing. V1 area was determined by CO staining as described (*Savier et al., 2017*; *Zembrzycki et al., 2015*) and shown on *Figure 3C*, lower left panel. All animal procedures were in accordance with national (council directive 87/848, October 1987), European community (2010/63/EU) guidelines and University Health Network. Official agreement number for animal experimentation is A67-395, protocol number 01831.01 and AUP 6066.4 (M.R).

### In silico simulations of the retino- and cortico-collicular maps

In the 3-step map alignment model, Ephas/Efnas forward signaling level is modeled by experimentally measured and estimated values of graded expression levels of Epha receptors (Epha3/4/5/6/7) and Efna ligands (Efna2/3/5) in RGCs, SC and V1 cortex, as previously described (*Tsigankov and Koulakov, 2006*; *Savier et al., 2017*). Competition, where two RGCs/V1 neurons cannot project to the same target in the SC, is modeled by indexation of 100 RGCs/V1 neurons projecting to 100 positions in the SC (*Koulakov and Tsigankov, 2004*; *Tsigankov and Koulakov, 2006*; *Savier et al., 2017*). Contribution of correlated neuronal activity -assuming Hebbian plasticity between RGC/V1 axons and collicular neurons in the SC is modeled by pair-wise attraction inversely proportional to the distance between two RGC/V1 neurons (*Savier et al., 2017*). The 3-step map alignment algorithm first simulates the projection of 100 RGCs along the nasal-temporal axis of the retina mapping onto the rostral-caudal axis of the SC (the RC map). Second, according to the layout of this map, retinal Efna gradients are transposed along the rostral-caudal axis of the SC. Third, the projections of

100 V1 neurons following the medial-lateral axis of V1 cortex are simulated onto the rostral-caudal axis of the SC. Forward Ephas/Efnas signaling is applied for both RC and CC projections.

The 3-step map alignment model in MATLAB (*Savier et al., 2017*) was used to simulate the formation of both the RC and CC map in the presence of an oscillatory Epha gradient in the retina. Each brain structure (retina, SC, V1) is modeled as a 1-d array of 100 neurons (N) in each network. Two maps are generated: first, the map from retina to SC; second, the map from V1 to SC. Each map is modeled sequentially in the same way. This model consists in the minimization of affinity potential (E) which is computed as follows:

$$E = E_{act} + E_{chem}$$

At each step, this potential is minimized by switching two randomly chosen axons probabilistically to reduces the energy in the system by Delta E ($\Delta E$). The probability of switching, p, is given by:

$$p = \frac{1}{(1 + e^{(4\Delta E)})}$$

$E_{chem}$ is expressed as follows:

$$E_{chem} = \sum_{i \in synapses} \alpha (R_A(i) - R_A(j))(L_A(i') - L_A(j'))$$

where $\alpha = 200$ is the strength, $R_A(i)$ and $R_A(j)$ the Epha receptor concentration in the retina and V1 at location (i) and (j) and $L_A(i')$ and $L_A(j')$ the Efna ligand concentration at the corresponding position (i') and (j') in the SC.

The contribution of activity-dependent process is modeled as:

$$E_{act} = -\frac{\gamma}{2} \sum_{i,j \in synapses} C_{ij} U(r)$$

where $\gamma = 1$ is the strength parameter, $C_{ij}$ is the cross-correlation of neuronal activity between two RGCs (or V1) neurons during spontaneous activity located in (i) and (j), and U simulates the overlap between two SC cells. Here, we use $C_{ij} = e^{\frac{-r}{R}}$, where r is the retinal distance between RGC (i) and (j), R = b x N, with b=0.11 and $U(r') = e^{\frac{-r'^2}{2d^2}}$, where r' is the distance between two SC points (i', j'), d = 3 and N = 1 to 100 neurons. Parameters of the model are presented *Table 1*.

## Gradients of ligands and receptors

### Retinal Epha gradients

Measured gradients of Epha receptors ($R_{Epha}$) in RGCs along the nasal-temporal axis (x) $R_{Epha}(x)^{retina}$ (*Figure 2A,G,M*, *Brown et al., 2000*; *Reber et al., 2004*) are modeled by two equations, one corresponding to Isl2(+) RGCs expressing WT levels of Ephas + Epha3 and the second corresponding to Isl2-negative (Isl2-) RGCs expressing the WT level of Ephas only:

$$
\begin{aligned}
R_{Epha\ Isl2+}(X)^{retina} &= R_{Epha5}(X) + R_{Epha6}(X) + R_{Epha4} + R_{Epha3} \\
&= 0.14e^{0.018x} + 0.09e^{0.029x} + 1.05 + R_{Epha3} \\
R_{Epha\ Isl2+}(X)^{retina} &= R_{Epha5}(X) + R_{Epha6}(X) + R_{Epha4} \\
&= 0.14e^{0.018x} + 0.09e^{0.029x} + 1.05
\end{aligned}
$$

With $R_{Epha3} = 0.93$ in Isl2$^{Epha3/+}$ and $R_{Epha3} = 1.86$ in Isl2$^{Epha3/Epha3}$, since Epha3 expression level depend on the number of copies of the knocked-in allele (*Brown et al., 2000*; *Reber et al., 2004*). The oscillatory gradient was generated by randomly attributing to 50% of collicular TZs an overexpression of Epha3 in a genotype-dependent manner.

### Collicular Efna gradients

Estimated gradients of Efna ligands ($L_{Efna}$) in the SC along the rostral-caudal axis (x), $L_{Efna}(x)^{SC}$ (*Figure 2B*, *Cang et al., 2005*; *Savier et al., 2017*; *Tsigankov and Koulakov, 2010*; *Tsigankov and Koulakov, 2006*):

$$L_{Efna}(X)^{SC} = e^{(x-100/100)} - e^{(-x-100/100)}$$

## Cortical V1 Epha gradients

Estimated gradient of Epha receptors ($R_{Epha}$) in V1 along the medial-lateral axis (x), $R_{Epha}$ (x) $^{V1}$ (*Tsigankov and Koulakov, 2010*; *Tsigankov and Koulakov, 2006*) for all genotypes:

$$R_{Epha}(X)^{V1} = e^{(-x/100)} - e^{(x-200/100)+1}$$

## Retinal Efna gradients

Measured gradients of Efna ligands in RGCs along the nasal-temporal axis $L_{Efna}$ (x)$^{retina}$ (*Savier et al., 2017*) for all genotypes:

$$L_{Efna}(X)^{retina} = L_{Efna5}(X)^{retina} + L_{Efna2}(X)^{retina} + L_{Efna3}(X)^{retina} = 1.79\,e^{-0.014x} + 1.85\,e^{-0.008x} + 0.44$$

## Retinal Efna gradients transposition

When transposed along the rostral-caudal axis in the SC (retina -> SC), retinal Efna gradients are flipped along the x axis and become:

$L_A(x)^{retina\ ->\ SC} = (1/1.78)\,e^{(-0.014(100-x))} + (1/1.85)\,e^{(-0.008(100-x))} + 0.44$
$L_A(x)^{retina\ ->\ SC} = 0.56\,e^{(0.014x)} + 0.54\,e^{(0.008x)} + 0.44$
Intrinsic Dispersion Index (IDI) – Local intrinsic Dispersion variation (IDV) - Alignment Index (AI).

The intrinsic dispersion index and local intrinsic variation, corresponding to the degree of dispersion of the maps are calculated by:

$$IDI = \left(\sqrt{\sum_{x=1}^{100}(y(x) - \bar{y})^2}\right)/100 \text{ (specific to each retino- and cortico-collicular map)}.$$

$$Local\ IDV = \sqrt{\sum_{x=1}^{100}(y(x+1) - y(x))^2} \text{ (specific to each retino- and cortico-collicular map)}$$

The alignment index is calculated by:

$$AI = \frac{1}{100}\sum_{x=1}^{100}|y_{x\ retino}$$

## Statistical analysis

Comparison of relative Efna/Epha mRNA expression levels in SC and V1 in Isl2$^{Epha3/Epha3}$ animals relative to WT were performed by one sample t-test, reference value = 1 as described in *Savier et al., 2017*. Comparison of relative Efnas expression levels, relative to WT Nasal in Nasal, Central and Temporal WT and Isl2$^{Epha3/Epha3}$ were performed by the two-way ANOVA without replication (Efna x genotype comparison). Experimental and simulated visual maps were compared using Kolmogorov-Smirnov 2-sample test.

Descriptive analysis of IDV curves was performed using:

- the Jaccard similarity index (J), which indicates the degree of similarity of two sets of data, varies between J = 0 (no similarity) and J = 1 (perfect similarity) and is calculated as follows (*Metcalf and Casey, 2016*):

$$J_{(X,Y)} = (|X \cap Y|/|X \cup Y|)$$

- Covariance Cov (x, y):

$$Cov(x,y) = \left(\sum(x_i - \bar{x})(y_i - \bar{y})\right)/N$$

where Cov>0 indicates similar trend and Cov <0 indicates opposite trend.

## Data availability

Leave-one-out 'Leave-one-out' script in R: https://github.com/michaelreber/Leave-one-out-LOESS/blob/master/wtloess.R (*Savier, 2020a*; copy archived at https://github.com/elifesciences-publications/Leave-one-out-LOESS).

3-step map alignment model 3-step map alignment script in MATLAB: https://github.com/michaelreber/3-step-Map-Aligment-Model (*Savier, 2020b*; copy archived at https://github.com/elifesciences-publications/3-step-Map-Aligment-Model-2020).

## Acknowledgements

We are grateful to Amelie Barthelemy and Frank Pfrieger, CNRS UPR3212 Strasbourg France for technical advice on retinal ganglion cell isolation. This work was supported by USIAS - University of Strasbourg, CNRS and start-up funds from the DKJ Eye Institute, Krembil Research Institute, University Health Network.

## Additional information

### Funding

| Funder | Grant reference number | Author |
| --- | --- | --- |
| University Health Network | Start-up grant | Michael Reber |
| Centre National de la Recherche Scientifique | Project grant | Michael Reber |
| Université de Strasbourg | Project grant - USIAS | Michael Reber |

The funders had no role in study design, data collection and interpretation, or the decision to submit the work for publication.

### Author contributions

Elise Laura Savier, Conceptualization, Resources, Data curation, Software, Formal analysis, Validation, Investigation, Visualization, Methodology, Writing - review and editing; James Dunbar, Investigation, Methodology, Writing - review and editing; Kyle Cheung, Investigation, Methodology; Michael Reber, Conceptualization, Resources, Data curation, Software, Formal analysis, Supervision, Funding acquisition, Validation, Investigation, Visualization, Methodology, Writing - original draft, Project administration, Writing - review and editing

### Author ORCIDs

Elise Laura Savier [iD] https://orcid.org/0000-0001-7512-1630
Michael Reber [iD] https://orcid.org/0000-0001-8842-7276

### Ethics

Animal experimentation: This study was performed in strict accordance with the recommendations in the Guide for the Care and Use of Laboratory Animals of University Health Network. All of the animals were handled according to approved institutional animal care and use committee (IACUC) protocols (#6066) of the University Health Network. All surgery was performed under sodium pentobarbital anesthesia, and every effort was made to minimize suffering.

### Decision letter and Author response

Decision letter https://doi.org/10.7554/eLife.59754.sa1
Author response https://doi.org/10.7554/eLife.59754.sa2

## Additional files

**Supplementary files**
• Transparent reporting form

**Data availability**

Codes are available on GitHub repository https://github.com/michaelreber/Leave-one-out-LOESS/blob/master/wtloess.R (copy archived at https://github.com/elifesciences-publications/Leave-one-out-LOESS) https://github.com/michaelreber/3-step-Map-Aligment-Model (copy archived at https://github.com/elifesciences-publications/3-step-Map-Aligment-Model-2020).

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
