## [Decision Letter]

Thank you for submitting your article "New insights on the modeling of the molecular mechanisms underlying neural maps alignment in the midbrain" for consideration by *eLife*. Your article has been reviewed by two peer reviewers, and the evaluation has been overseen by a Reviewing Editor and Andrew King as the Senior Editor. The following individual involved in review of your submission has agreed to reveal their identity: Xavier Nicol.

The reviewers have discussed the reviews with one another and the Reviewing Editor has drafted this decision to help you prepare a revised submission.

Summary:

The reviewers appreciated this follow-up to your 2017 *eLife* paper investigating the relationship between retinocollicular and cortico-collicular map development. Using your 3-step map alignment that you applied to the Isl2-Epha3KI in the 2017 paper, you now further evaluate the robustness of the 3-step map alignment in another mouse line with visual map duplication, the Isl2::EphA3 line. You nicely show that your 3-step map alignment algorithm accurately simulates both retino- and cortico-collicular mapping defects in the Isl2-Epha3KI, the originally used genetic model for this issue in the Brown et al. and Triplett et al. studies. You indicate that the mechanism underlying the subsequent duplication of the cortico-collicular projection corresponds to a redistribution of retinal molecular cues, the ephrin As (Efnas), into the SC. You implicate that Efnas provided by the retinal axons projecting within the SC act together with correlated activity to instruct cortico-collicular alignment onto the retinocollicular map. Your results, carefully executed with both modeling and experimental analysis, as you state, "challenge previous hypotheses and provide an alternative, although complementary, explanation for the phenotype observed".

The study falls within the guidelines for a Research Advance, "a short article that allows either the authors of an *eLife* paper or other researchers to publish new results that build on the original research paper in an important way." The reviewers and I thought that your study certainly further validates the 3-step map alignment computational model.

There are a number of edits and additions that the reviewers called for:

Essential revisions:

You pose the idea that correlated activity plays is a driving force in matching cortico-collicular maps to retinocollicular maps. You also use several novel quantification methods to assess map alignment and the organization of the maps. These quantification methods seem quite powerful for assessing map alignment differences (compare Figure 4E, F with Alignment Index in Figure 4G). You state that your "data confirm the validity and robustness of our algorithm". Additional points to mention include:

1) The data presented here apply this model only to the Isl2-EphA3 Ki/Ki and Isl2-Efna3 Ki/Ki models. Since the justification for generating this model was that other models cannot faithfully "reproduce all nuances of experimental findings", it seems that this model should also have been compared to the other experimental models specifically referenced: EfnaKOs and Math5KO. Such an analysis is probably beyond the scope of your present study but nonetheless bears mention in the text as a future direction.

2) You seek to make the point that "transposed" retinal Efnas (most likely Efna3 based on the data from the previous study) play a more significant role in cortico-collicular map alignment than correlated activity. This remains to be demonstrated experimentally by removing retinal derived Efna3 (or other Efnas) and assessing cortico-collicular maps (in the context of a WT SC or an Isl2-EphA3 Ki/Ki SC). This would call for a separate study but the need for such a study should be acknowledged in the Discussion.

3) You mention that correlated activity is taken into account in the 3-step model without further explanation.

a) Clarification of this point is needed, especially in the framework of the discussion that proposes an alternative explanation of the map alignment phenotype in Isl2::EphA3 knock-in animals.

b) Retinothalamic mapping is likely to be affected, at least in EphA3 knock-in animals, but likely not in ephrinA3 knock-in animals according to your results. How would this reorganization of the dLGN input and in turn in V1 electrical activity impact the retino- and cortico-collicular map alignment? Mention of this point would also be welcome in the Discussion.

4) Based on the model you proposed in your 2017 paper for retina-derived Efna3 in cortico-collicular mapping, it is not clear why the cortico-collicular maps are more misaligned rostrally than caudally in Isl2-Efna3Ki/Ki mutants (see Figures 4F and 5E), and why this is essentially opposite to the misaligned maps in the Isl2-Eph3AKi/+ or Isl2-Eph3AKi/Ki mutants). Please comment in the Discussion.

5) You quantify expression levels of ephrinAs and EphAs in the superior colliculus and in V1 in Isl2::EphA3 animals. Since no difference was detected with controls, a positive control would further enhance the quality of the dataset. Do you have quantitative data on the expression of EphA3 in the retina?

---

## [Author Response]

Essential revisions:You pose the idea that correlated activity plays is a driving force in matching cortico-collicular maps to retinocollicular maps. You also use several novel quantification methods to assess map alignment and the organization of the maps. These quantification methods seem quite powerful for assessing map alignment differences (compare Figure 4E, F with Alignment Index in Figure 4G). You state that your "data confirm the validity and robustness of our algorithm". Additional points to mention include:1) The data presented here apply this model only to the Isl2-EphA3 Ki/Ki and Isl2-Efna3 Ki/Ki models. Since the justification for generating this model was that other models cannot faithfully "reproduce all nuances of experimental findings", it seems that this model should also have been compared to the other experimental models specifically referenced: EfnaKOs and Math5KO. Such an analysis is probably beyond the scope of your present study but nonetheless bears mention in the text as a future direction.

We agree with the reviewer’s comment and we have now addressed this point in our revised manuscript, as follows:

“Further validation of the 3-step alignment model would involve testing this algorithm using other genetic models such as the Math5KO, which presents an unrefined cortico-collicular map. The prediction of the cortico-collicular map phenotype of compound mutants such as Efnas triple knock-out and EphA4KO crossed with EphA3KI would also be of prime interest but would require the experimental assessment of the cortico-collicular map in these mutants, which to our knowledge, has not been performed to this date.”

2) You seek to make the point that "transposed" retinal Efnas (most likely Efna3 based on the data from the previous study) play a more significant role in cortico-collicular map alignment than correlated activity. This remains to be demonstrated experimentally by removing retinal derived Efna3 (or other Efnas) and assessing cortico-collicular maps (in the context of a WT SC or an Isl2-EphA3 Ki/Ki SC). This would call for a separate study but the need for such a study should be acknowledged in the Discussion.

The generation of animals carrying an RCGs-specific ablation of Efna3 is of prime interest, though beyond the scope of this study. We hope to be able to conduct such experiments in the near future (as well as addressing the role of each Efna in the retina) together with the manipulation of correlated activity. We acknowledge the need for this experimental demonstration in the Discussion, as follows:

“The 3-step map alignment model suggests that retinal EfnAs are responsible for the cortico-collicular map duplication in the Isl2-EphA3KI mutants. An additional validation of this mechanism would require the mapping of Isl2-EphA3KI cortico-collicular map in compound mutants in which Efnas have been specifically removed in the retina (Suetterlin et al., 2014). The absence of duplication in the cortico-collicular map in such a mutant would further demonstrate the implication of retinal EfnAs in the alignment of visual maps in the SC.”

3) You mention that correlated activity is taken into account in the 3-step model without further explanation.a) Clarification of this point is needed, especially in the framework of the discussion that proposes an alternative explanation of the map alignment phenotype in Isl2::EphA3 knock-in animals.

The implementation of correlated activity is mentioned in the Materials and methods section, “Contribution of correlated neuronal activity -assuming Hebbian plasticity between RGC/V1 axons and collicular neurons in the SC is modeled by pair-wise attraction inversely proportional to the distance between two RGC/V1 neurons” and further details can be found in the subsection “In silico simulations of the retino- and cortico-collicular maps”. However, for clarification, we have added further details in the Results section, “Simulation of the Isl2-Epha3KI mutants retino- and cortico-collicular mapping”, as follows: “while correlated activity attracts projections originating from similar locations activity through pair-wise attraction, which is inversely proportional to the distance between two RGC/V1 neurons.”

Our model suggests that molecular cues originating from the retina are contributing to the duplication and are further strengthen by the correlated activity driven by the duplicated RC projection. Both forces act together, leading to a phenotype with 100% penetrance in the Isl2-EphA3KI. These results contrast with the more variable phenotype of the Isl2-EfnA3, where molecular cues and correlated activity act against each other (see Discussion, subsection “The basic map organization principle encoded in the 3-step map alignment model provides an alternative explanation for map alignment defects in Isl2-Epha3KI animals”).

b) Retinothalamic mapping is likely to be affected, at least in EphA3 knock-in animals, but likely not in ephrinA3 knock-in animals according to your results. How would this reorganization of the dLGN input and in turn in V1 electrical activity impact the retino- and cortico-collicular map alignment? Mention of this point would also be welcome in the Discussion.

Previous studies have shown that the retino-thalamic projections in Isl2-EphA3 knock-in are not duplicated (see Triplett, 2009, Supplementary Figure 2. “The retinogeniculate and geniculocortical projections are not anatomically duplicated in EphA3ki/ki mice”) and that the cortical map in V1 does not display abnormalities (see Triplett et al., 2009 Figure 2, “EphA3^ki/ki^ Mice Have Duplicated Functional Maps in the SC and a Single Functional Map in V1”).

4) Based on the model you proposed in your 2017 paper for retina-derived Efna3 in cortico-collicular mapping, it is not clear why the cortico-collicular maps are more misaligned rostrally than caudally in Isl2-Efna3Ki/Ki mutants (see Figures 4F and 5E), and why this is essentially opposite to the misaligned maps in the Isl2-Eph3AKi/+ or Isl2-Eph3AKi/Ki mutants). Please comment in the Discussion.

The total retinal Efna gradients transferred to the SC in Isl2^Efna3/Efna3^ are oscillatory, running from high-caudal to low-rostral. Based on the mechanisms of relative Epha/Efna signaling demonstrated by Brown and collaborators (Brown et al., 2000), we speculate that the relative difference in cortical EphA/transferred retinal Efna signaling in the caudal SC is lower than in the rostral SC. In other words, the relative difference of transferred Efnas between Isl2(+) RGC axons and Isl2(–) RCG axons is lower in the caudal SC due to the overall high Efna level (relative difference = DC/C_total_). Whereas in the rostral SC, the overall Efna level is low and therefore the relative difference (DC/C_total_) between Isl2(+) and Isl2(-) RGCs is larger leading to a more prominent effect on cortico-collicular TZ duplications in the rostral SC compared to caudal SC in Isl2^Efna3/Efna3^ animals. Moreover, the correlated activity of the normal RC map in the Isl2^Efna3/Efna3^ animals may tend to limit the effect of the oscillatory gradients on cortico-collicular duplication, however, a normal RC activity in the Isl2^Efna3/Efna3^ has yet to be formally confirmed.

In Isl2^Epha3/Epha3^ animals, the retinal Efna gradients are similar to WT. The duplication of the cortico-collicular mapping is a consequence of both duplicated retinal Efna gradients transferred into the SC (due to duplicated RC map) *and* duplicated correlated activity (Triplett et al., 2009). In the Isl2^Epha3/Epha3^ there is an overall duplication of the transferred retinal Efna gradients (two contiguous exponential gradients) whereas in the Isl2^Efna3/Efna3^ there is a local oscillatory gradient generating local duplication of cortico-collicular projections (Figure 9 in Savier et al., 2017; Figure 1 in Savier and Reber, 2018).

5) You quantify expression levels of ephrinAs and EphAs in the superior colliculus and in V1 in Isl2::EphA3 animals. Since no difference was detected with controls, a positive control would further enhance the quality of the dataset. Do you have quantitative data on the expression of EphA3 in the retina?

In WT animals, EphA3 was shown to be strongly expressed in the retina but NOT in RGCs (Brown et al., 2000, Figure 1A). As requested by the reviewers, we performed a qRT-PCR to measure EphA3 expression in Isl2^Epha3/Epha3^ animals on isolated RGCs from nasal, central and temporal P1/2 retina. Epha3 expression in WT RGCs cannot be detected. In Isl2^Epha3/Epha3^ animals, expression of EphA3 is detected in RGCs and shows homogenous expression between nasal, central and temporal retinas, in line with previous data (Reber et al., 2004 Figure 2A). This result has been added to Figure 1A.